# The Impact of Mountain Range Geographic Orientation on the Altitude Effect of Precipitation $\delta^{18}$O in the Upper Reaches of the Heihe River Basin in the Qilian Mountains

**Jianqiao He \*, Wei Zhang and Yuwei Wu**

State Key Laboratory of Cryospheric Science, Northwest Institute of Eco-Environment and Resources, Chinese Academy of Sciences, Lanzhou 730000, China; zhangw06@lzb.ac.cn (W.Z.); htwyw@lzb.ac.cn (Y.W.)

\* Correspondence: hejqcn@lzb.ac.cn; Tel.: +86-093-149-67372

**Abstract:** The precipitation $\delta^{18}$O-elevation gradients are important for paleoclimate, hydrology, and paleoelevation studies. The field setting for this research was the upper reaches of the Heihe River Basin within the Qilian Mountains in the Northern Tibetan Plateau. Three study sites were established along the Heihe main river. These sites were the Yingluoxia and Qilian hydrological stations and the Yeniugou meteorological station. The Yingluoxia hydrological station was the dividing point between the upper and middle reaches of the Heihe River Basin. The altitudes of these sites range from 1600 m to 3300 m. Summer precipitation is predominant with regard to the annual precipitation amount. By analysis of variance (ANOVA), the precipitation $\delta^{18}$O data collected from the three sites were analyzed, spanning a year of precipitation data from 2007.10 to 2008.9. The results showed that the $\delta^{18}$O-elevation gradient was not significant ($\alpha = 0.05$) at a seasonal or annual scale in this region and the precipitation-weighted annual mean $\delta^{18}$O was −7.1‰. Mechanisms that have been proposed to explain this result consider the role of two processes, including (1) mixing of moisture sources, a process common in an arid and semiarid region, and (2) the absence of a mechanism for water vapor to climb along slopes in the precipitation system. Atmospheric water vapor mainly travels along the trend of the Qilian Mountains range rather than climbing it because this region is dominated by the westerlies and the trend of the Qilian mountains is geographically aligned to the NWW (north-west-west) direction. We demonstrated that, aside from the water vapor source, the spatial relationship between the water vapor transport pathway and the trend of the mountain range are the main driving factors associated with the stable isotope trends in precipitation. As a result, it is important to re-recognize the timing and location of groundwater recharge in the Heihe River Basin.

**Keywords:** precipitation; $\delta^{18}$O; altitude effect; ANOVA; trend of the Qilian mountains

## 1. Introduction

The continental interior (e.g., the western United States and the Tibetan Plateau) is the focus region for many paleoaltimetry studies. The relationship between elevation and isotopic composition has been of particular interest to the field of quantitative paleoaltimetry; it has garnered significant attention in recent decades due to the importance of accurate elevation histories for constraining the geodynamic, tectonic, and climatic evolution of major orogenic systems [1]. Paleoaltimetry reconstructions often utilize isotopic relationships between modern precipitation and elevation to constrain paleoelevation histories. In recent years, some researchers reconstructed the uplifting process of the Tibetan Plateau using modern precipitation isotopic proxies for paleoaltimetry [2–5]. This approach exploits the altitude effect, which is the observation that the stable isotopic composition of meteoric water becomes

more depleted at higher elevations. Rowley et al. [2] presented oxygen-isotope-based estimates of the paleoaltimetry of the late Eocene and younger deposits of the Lunpola Basin in the central plateau, which indicated that the surface of Tibet had been at an elevation of more than 4 kilometers for at least the past 35 million years. Apart from requiring authentic paleo-meteoric water proxies, the accurate precipitation $\delta^{18}$O-elevation gradients play a key role in this study. Lacustrine carbonates from the Eocene –Oligocene Fenghuoshan Group of the North-Central Tibetan Plateau showed that the $\delta^{18}$O$_{VSMOW}$ values of regional Eocene–Oligocene meteoric waters had an average composition of $-9.7 \pm 2.8$‰. Using $\delta^{18}$O-elevation gradients ($-0.2$–$0.3$‰ /100 m) established in the distant Nepalese Himalayas, an Eocene paleoelevation of 2000 m has been proposed for this region [4]. In contrast, the application of the modern precipitation $\delta^{18}$O-elevation gradient of Northern Tibet in this region ($-0.15$‰ /100 m) yields paleoelevation estimates that are indistinguishable from modern Tibetan Plateau elevations (4500 m–5000 m) [6,7].

The precipitation $\delta^{18}$O-elevation gradient has been used widely in groundwater recharge studies. In a study of recharge to a geothemermal system at Mount Meager (a quaternary volcano in the Coast Range of Western British Columbia), precipitation collected from 11 sites between altitudes of 250 m and 3250 m showed a $\delta^{18}$O-elevation gradient of $-0.25$‰/100 m, which provided evidence of the recharge environment of the thermal groundwaters [8]. Chen Jiansheng et al. [9] studied the deep confined water recharge environment in the lower reaches of the Heihe River in the Ejin Basin area using environmental isotope-elevation gradients, temperature, and electric conductivity. They concluded that the confined groundwater recharge in the Ejin Basin was derived from precipitation over the Qilian Mountains.

Making accurate estimates of precipitation isotope-elevation gradients is important for paleoaltimetry reconstructions, distinguishing groundwater recharge environments, and identifying the major generation area of water resources in arid regions. Due to differences in regional atmospheric circulation systems, the $\delta^{18}$O-elevation gradients of global precipitation range from $-0.10$ to $-1.1$‰/100 m [8], with an average of $-0.28$‰/100 m [1]. In the vast Tibetan Plateau, there are significant differences in the water vapor sources and water circulation patterns in different regions; therefore, the processes and mechanisms that influence moisture source and water cycling are significantly different in different areas, and thus, it is necessary to study the characteristics of altitude effects of stable isotopes in different regions of the Tibetan Plateau. Some studies have investigated the altitude effects of stable isotopes of water in specific regions of the Tibetan Plateau. Some researchers have quantified the value of $\delta^{18}$O-elevation gradients of precipitation and river water in the Southeast Tibetan Plateau and the Southern Himalayas [10,11]. Ding Lin et al. [12] built an empirical model to examine the relationship between the elevation and concentration of oxygen isotopes in river water in the Tibetan Plateau. Stable isotope tracing techniques have been extensively applied to studies of the formation and transformation of water resources and water cycling in the Qilian Mountains of the Northern Tibetan Plateau [13–19]. In some of these studies, the relationship between $\delta^{18}$O and elevation was estimated by simple linear regression with no further statistical analysis of the data. This approach is insufficient for the development of the research fields mentioned above in the Tibetan Plateau.

In this study, we measured $\delta^{18}$O in precipitation collected from three sites located across a topographic gradient in the upper reaches of the Heihe River Basin in the Qilian Mountains. The relationship between precipitation $\delta^{18}$O and elevation was investigated by analysis of variance. This study provides additional support for the research of paleoaltimetry reconstructions and water cycling within the Tibetan Plateau.

## 2. Study Area and Precipitation Sampling

### 2.1. Site Description

The Qilian Mountains, mainly dominated by a series of parallel NWW ridges and valleys, are located in the Northern Tibetan Plateau and their slopes dip to the south and north [20,21]. The study region was located in the middle of the Qilian mountains within an elevation range of 1600 to 5100 m a.s.l. (above sea level), see Figure 1. Steep valleys have dense vegetation cover in the study area, rain falls copiously, and annual precipitation amounts exceed 200 mm and can even exceed 600 mm in some alpine regions [22–24]. The Qilian Mountains are the water source area for the Heihe River Basin, the second largest inland river basin in China. In the Heihe River Basin, the proportion of precipitation to the mountainous runoff was estimated to be 90% [25]. The precipitation in the Qilian Mountains makes up the majority of the water available for irrigation and municipal water supplies, providing water for over 1.5 million people living in the middle and lower reaches. The study of hydrological processes in this region has important significance for science and society [26,27].

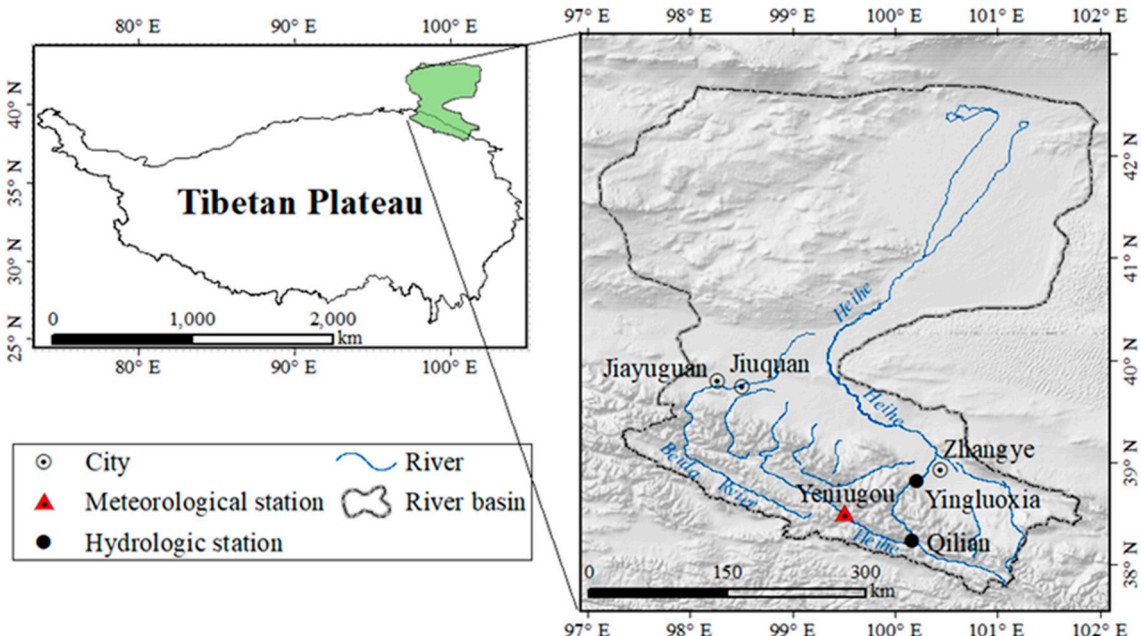

**Figure 1.** The Heihe River Basin and sampling site locations.

### 2.2. Precipitation Sampling

Three sampling sites were established along the main Heihe River, see Figure 1. One site was the Yeniugou meteorological station (3320 m a.s.l.), which is located within the source area of the Heihe River. The other two sites were located at the Qilian hydrological stations (2787 m a.s.l.) and Yingluoxia hydrological station (1674 m a.s.l), which is the dividing point between the upper and middle reaches of the Heihe River Basin. The spatial range of the sampling locations covers the upper reaches of the Heihe River Basin.

Since the error from the sampling process is greater than the error from the sample analysis [28], a detailed sampling procedure was established. Precipitation samples were collected for each precipitation event at the three stations between 2007.10 and 2009.9. The air temperature and humidity at the beginning and end of the precipitation event were recorded, as well as the total rainfall. When the rain cleared, the samples were packed in 30 mL PET (Polyethylene terephthalate) plastic bottles to be protected from evaporation.

The Yeniugou meteorological station samples between 2008.10 and 2009.9 were omitted, so the precipitation $\delta^{18}$O-elevation gradient discussion focuses only on isotopic data collected from 2007.10

to 2008.9 (one full year). Isotopic measurements were completed at the State Key Laboratory of Cryospheric Sciences, Chinese Academy of Sciences. Water isotope ($\delta$D, $\delta^{18}$O) values were measured via a Picarro liquid-water isotope analyzer, with accuracies of 0.5 ‰ for $\delta$D and 0.025‰ for $\delta^{18}$O. All of the isotope values are expressed in per mil notation (‰) relative to the Vienna standard mean ocean water (VSMOW). The precipitation-weighted annual and season mean $\delta^{18}$O was calculated using Equation (1):

$$\delta^{18}O_{weighted} = \frac{\sum_1^n di \cdot pi}{\sum_1^n pi} \tag{1}$$

where *di* is each precipitation event's $\delta^{18}$O value, *pi* is the precipitation amount for that individual event, and *n* is the number of the precipitation events during the observation period.

## 3. Results

### 3.1. The Seasonal Variation of Precipitation $\delta^{18}O$

During the observing period (2007.10–2008.9), 58 precipitation samples were collected from the Yeniugou meteorological station and precipitation $\delta^{18}$O values ranged from −25.5‰–2.8‰ with an arithmetic mean of −9.1‰ and a precipitation-weighted mean $\delta^{18}$O of −6.7‰. The number of samples collected from the Qilian hydrological station was 103 and these had a $\delta^{18}$O value range of −21.8–4.7‰, an arithmetic mean of −7.4‰, and a precipitation-weighted mean of −7.2‰. For the Yingluoxia hydrological station, 49 precipitation samples were collected, and the $\delta^{18}$O values ranged from −25.9–6.0‰, with an arithmetic mean of −6.8‰ and a precipitation-weighted mean of −7.5‰.

Most annual precipitation fell during the warm season (2008.5–2008.9) in this region, as shown in Figure 2. The proportion of cold season precipitation (2007.10–2008.4) relative to the annual precipitation (2007.10–2008.9) was 21% at the Yingluoxia hydrological station, 12% at the Qilian hydrological station, and 11% at the Yeniugou meteorological station. Winter precipitation was much lower than summer precipitation. This result is similar to those of previous studies [15–17,29,30].

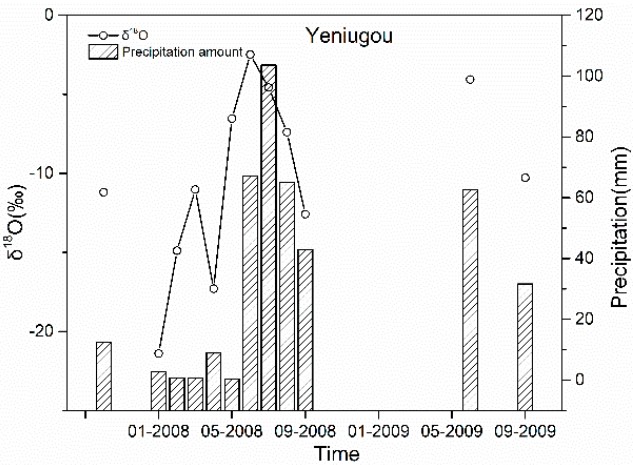

**Figure 2.** *Cont.*

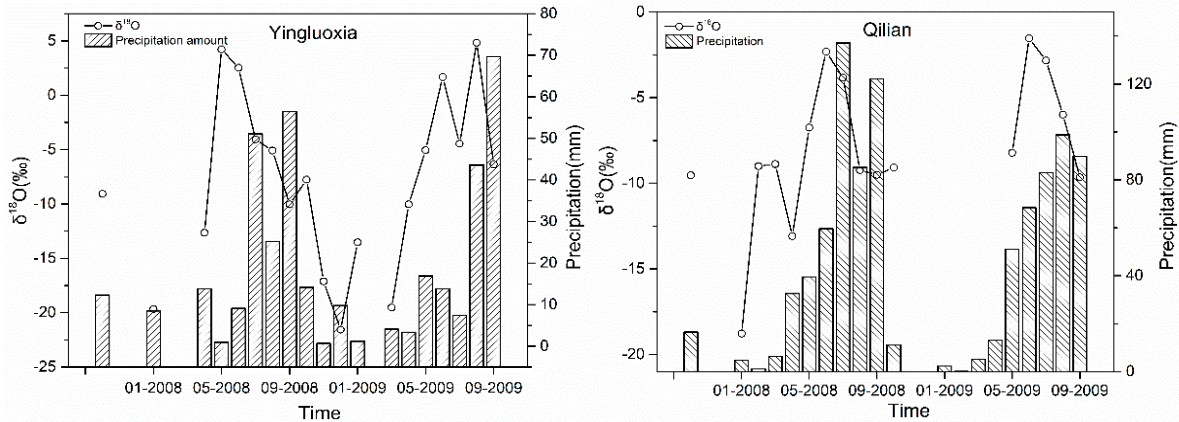

**Figure 2.** Time series of precipitation and $\delta^{18}O$ ratio of rainfall (2007.10–2009.9).

## 3.2. The Precipitation $\delta^{18}O$-Elevation Gradient

The altitude span between the Yeniugou meteorological station and the Yingluoxia hydrological station is 1600 m. That showed that the change of annual mean values of $\delta^{18}O$ in precipitation between the three stations was very small, see Figure 3. The difference between the maximum and minimum annual mean values was only 0.8‰, and the precipitation $\delta^{18}O$-elevation gradient was −0.04‰/100 m, see Figure 3b. Analyses of previous data on the isotopic content change in precipitation revealed a similar result, see Figure 3a [15]. During 2006.5–2007.5, the difference between the maximum annual mean and the minimum annual mean of precipitation $\delta^{18}O$ was only 1.3‰ at four stations (Yingluoxia, Qilian, Zhamashi hydrological station, and Yeniugou meteorological station). Precipitation-weighted mean $\delta^{18}O$ values hardly changed with elevation and the precipitation $\delta^{18}O$-elevation gradient was only −0.07‰/100 m. Zhang Yinghua et al. [16] measured the precipitation-weighted mean $\delta^{18}O$ in precipitation collected from fourteen sites in the upper reaches of the Heihe River Basin in the Qilian Mountains and they found a relationship between $\delta^{18}O$ in precipitation and elevation that they described by $\delta^{18}O = -0.0018H - 1.6891$ ($R^2 = 0.22$). Although the simple linear relationship indicated that the $\delta^{18}O$-elevation gradient was −0.18‰/100m, the regressions between precipitation-weighted mean $\delta^{18}O$ and elevation showed a weak correlation with an $R^2$ value of 0.22.

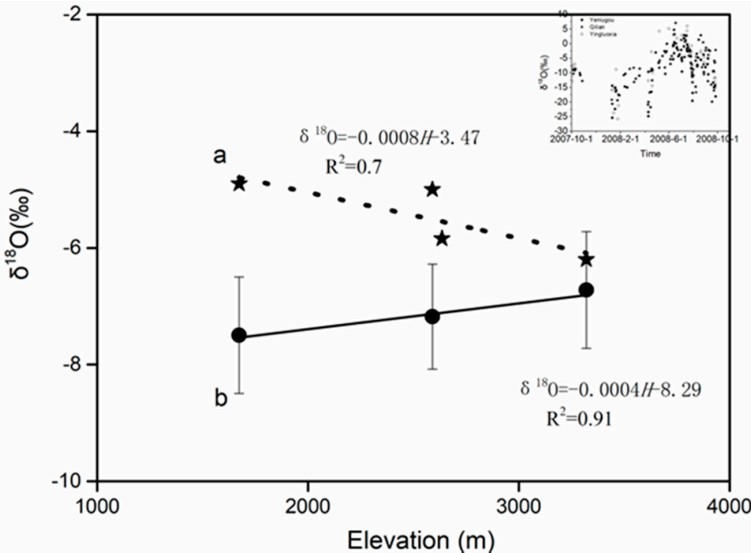

**Figure 3.** The $\delta^{18}O$ variability with elevation during sampling between 2007.10 and 2008.9 in the upper reaches of the Heihe River Basin in the Qilian Mountains. Error bars represent the relative standard deviation (RSD). The inset shows the $\delta^{18}O$ values for individual precipitation events in the three sampling sites. a: 2006.5–2007.5 (from Ninglian Wang et al. [18]); b: 2007.10–2008.9 (this study).

We found that the results of this study were only one-seventh to one-fourth of the $\delta^{18}$O-elevation gradients in other regions of the Tibetan plateau. For example, on the southern slope of the Himalayas, the average yearly precipitation-weighted mean $\delta^{18}$O-elevation gradient was measured to be $-0.15‰/100$ m [10] and the $\delta^{18}$O-elevation gradient was measured to be $-0.3‰/100$ m in the Southern Tibetan plateau and surrounding area [31]. Our study was smaller than the global average of $\delta^{18}$O-elevation gradients ($-0.28‰/100$ m), and exceeded the range of the global precipitation $\delta^{18}$O–elevation gradients ($-0.10$–$-1.1‰/100$ m [8]).

These results point to precipitation $\delta^{18}$O-elevation gradients being smaller in the Qilian Mountains. To assess whether or not the elevation variation has a significant influence on the $\delta^{18}$O of precipitation in the region, statistical analyses were performed by analysis of variance (ANOVA). ANOVA assesses whether or not the variation of a control variable has a significant influence on the observation variability though variance of an observation variability. We regarded the isotopic value of a single precipitation event as a sample element. The sample space was the set of all of the precipitation $\delta^{18}$O data for each of the three stations during the observation period (2007.10–2008.9). The sample capacities for the stations were 58 ($n1 = 58$) for the Yeniugou meteorological station, 103 ($n2 = 103$) for the Qilian hydrological station, and 49 ($n3 = 49$) for the Yingluoxia hydrological station. Data were analyzed by one-way ANOVA with SPSS 20(IBM), see Tables 1 and 2. Two hypotheses were established: The null hypothesis (H0), stated that the three population means were all the same, and the alternative hypothesis (H1), stated that the three population means were not the same or not all. An alpha level of 0.05 was used and, from the ANOVA report, F = 1.861 and P = $0.158 > \alpha = 0.05$. As a result, the null hypothesis (H0) was accepted. ANOVA revealed that the differences in the annual mean precipitation $\delta^{18}$O among the three points were not significant at a significance level of 5%. In other words, there was no significant correlation between elevation and annual mean precipitation $\delta^{18}$O in this study region.

**Table 1.** Test of homogeneity of variance.

| Levene Statistic | df1 | df2 | Sig. |
|:---:|:---:|:---:|:---:|
| 1.042 | 2 | 207 | 0.354 |

df1 = degrees of freedom between groups; df2= degrees of freedom within groups; Sig. = significance.

**Table 2.** One-way analysis of variance (ANOVA).

| | Sum of Squares | df | Mean Square | F | Sig. |
|:---|:---:|:---:|:---:|:---:|:---:|
| Between Groups | 171.397 | 2 | 85.698 | 1.861 | 0.158 |
| Within Groups | 9533.468 | 207 | 46.055 | | |
| Total | 9704.865 | 209 | | | |

df = degrees of freedom; F = F-test statistic; Sig. = significance.

More rainfall occurs during the warm season in the case study area. The rainfall-weighted means $\delta^{18}$O of the three sites during the warm and cold seasons from 2007.10 to 2008.9 are given in Figure 4. As shown in Figure 4, the linear correlation showed a low determination coefficient (warm season $R^2 = 0.02$, cold season $R^2 = 0.05$) between elevation, the mean values of precipitation $\delta^{18}$O in the warm and cold season are weak, and the precipitation $\delta^{18}$O-elevation gradient was very close to $0.00‰$. The result of ANOVA showed that there was a small difference between the $\delta^{18}$O mean values of these three stations in the warm and cold season, which amounted to no significant altitude effect. This result agrees with the analysis of the relationship between the annual precipitation $\delta^{18}$O mean values and elevation. In short, both at the annual and seasonal scale, the results showed that the altitude effect of $\delta^{18}$O in precipitation was not significant in this region.

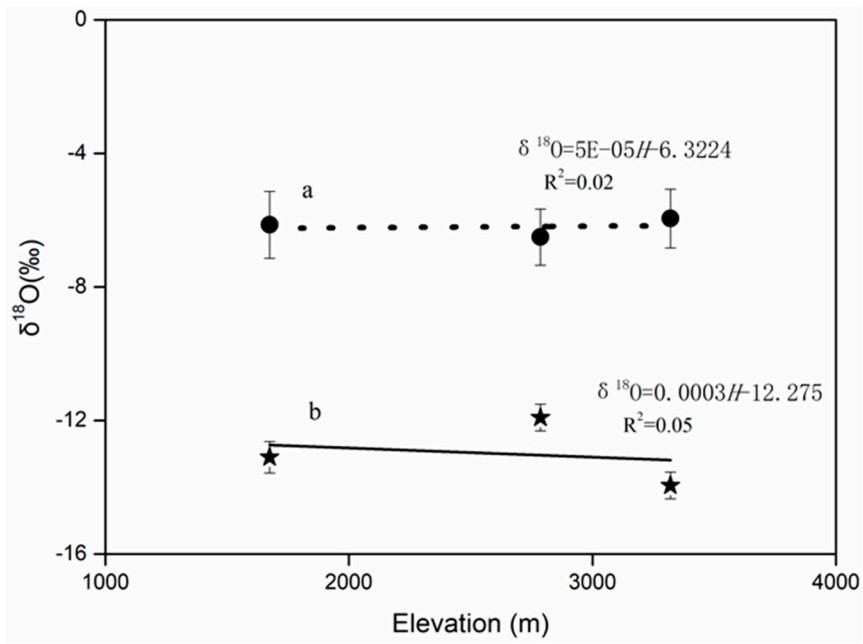

**Figure 4.** The $\delta^{18}O$ variability with elevation in the warm and cold seasons from 2007.10 to 2008.9 in the upper reaches of the Heihe River Basin. Error bars represent relative standard deviation (RSD). a: warm season (2008.5–2008.9); b: cold season (2007.10–2008.4).

*3.3. Trajectory Analysis of the Atmospheric Water Vapor*

Atmospheric water vapor is the material basis of precipitation, and the source and evolutionary process is reflected by the variation of precipitation $\delta^{18}O$. A NOAA Hysplit Trajectory Model was run for a 10-day backward trajectory analysis of individual precipitation events at three stations (Yeniugou, Qilian, and Yingluoxia) in the period from 2007.10 to 2008.9 to determine the origin of air masses, see Figure 5. The backward trajectory starting height was set to 2000 m above the ground, where the maximum moisture flux occurs [32]. Statistical analysis of the proportion of precipitation condensed by different moisture sources in the annual precipitation was calculated using the following equation:

$$k = \frac{\sum p}{\sum P} \tag{2}$$

where $\sum p$ is the sum of the precipitation amounts at the individual stations that were condensed by the same source of moisture. Moisture likely came from the west, north, or northwest if transported by the westerly belt, and from the east, south, or southeast if transported by monsoons. $\sum P$ is the sum of the total precipitation at the individual stations during the period from 2007.10 to 2008.9.

The simulation results showed that precipitation during the year at the three sites was mainly derived from westerly sources, though monsoon sources also contributed. The proportion of westerly precipitation relative to the total precipitation from 2007.10 to 2008.9 was 62.5% at Yingluoxia Station, 79.5% at Yeniugou Station, and 89.1% at Qilian Station, see Table 3.

Using the Hysplit Trajectory Model, air mass trajectories were simulated for each season from 2007.12 to 2008.11 above the Yingluoxia station. The results showed that almost all air masses came from the west or northwest during spring, autumn, and winter and that air masses were transported by monsoon during the summer with 9% of the air masses coming from the east, and 91% coming from the west. In short, the westerly belt appeared to be responsible for the transport of most water vapor in the region, in agreement with previous studies [17,29–31,33]. We did not distinguish the local water vapor from transported water vapor during the simulation. Instead, the source of water vapor was simply divided between the westerly belt or monsoon based on source assessment. The proportion

of rainfall formed by local water vapor in the annual precipitation in the region has been studied in more depth.

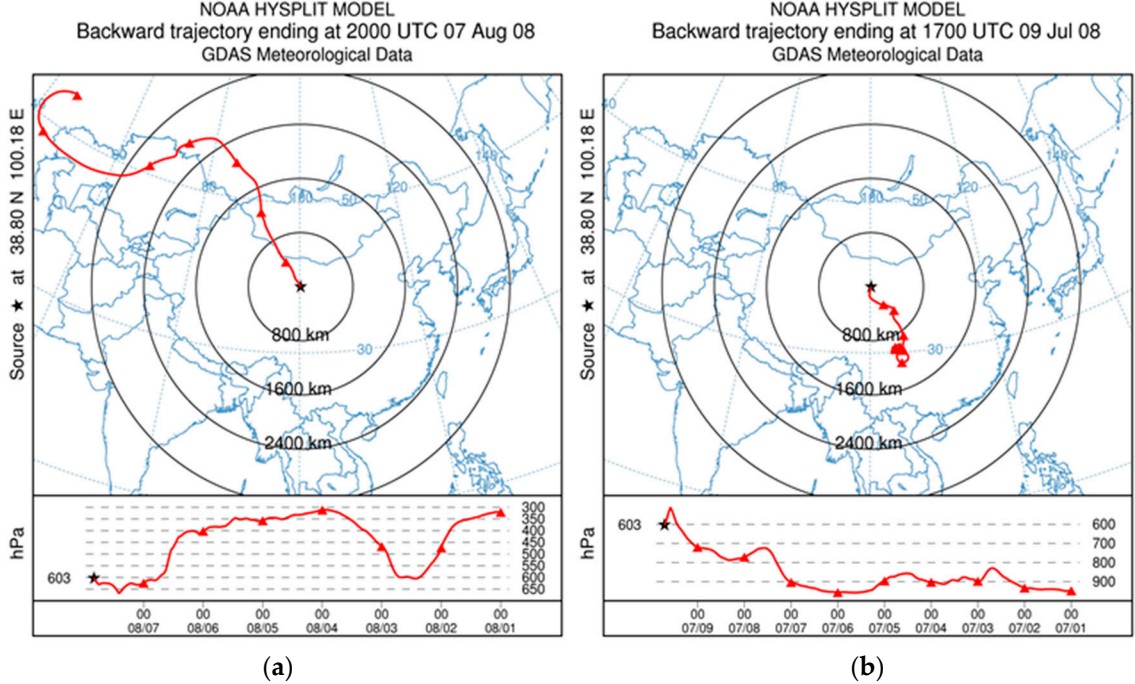

**Figure 5.** Two typical cases of air mass back trajectories of individual precipitation events by the Hysplit model for (**a**) 2008.8.7 at Yingluoxia station and (**b**) 2008.6.9 at Yingluoxia station.

**Table 3.** Analysis of precipitation from different water vapor sources in the upper reaches of the Heihe River Basin from 2007.10 to 2008.9. The number of precipitation samples from different sources for each site is given in the column labeled "n".

| Source of Moisture | Yingluoxia | | Qilian | | Yeniugou | |
|---|---|---|---|---|---|---|
| | n | Proportion in Total Rainfall during the Period (%) | n | Proportion in Total Rainfall during the Period (%) | n | Proportion in Total Rainfall during the Period (%) |
| Westerly belt | 37 | 62.5 | 71 | 89.1 | 41 | 79.5 |
| Monsoon | 12 | 37.5 | 32 | 10.9 | 16 | 20.5 |
| Total | 49 | 100 | 103 | 100 | 57 | 100 |

Due to the possibility of undersampling during the sampling process, the statistical value of precipitation should be smaller than the actual value.

## 4. Discussion

The isotopic composition of precipitation tends to change with the altitude of the terrain and becomes more and more depleted in $\delta^{18}O$ and $\delta D$ at higher elevations. This altitude effect of precipitation isotopic composition is best described by an ideal Rayleigh law in an open system. Condensation of rain is a phase separation process in an open system. As rain condenses, the heavier isotopes of water are preferentially removed from the air mass, and the air mass consequently becomes progressively lighter in isotopic composition along the moisture trajectory. The altitude effect on the stable isotope composition in modern precipitation also displays regional variations [8].

Djebou et al. [34] investigated the potential interactions between watershed characteristics and precipitation variability and the magnitude of watersheds' effects on precipitation was assessed. Their results show the necessity to include watershed topography in global and regional climate model parameterizations. This altitude effect is caused by increased rain at higher elevations due to continuous cooling of the air mass pseudo-adiabatically to below the dew point in an orographic precipitation system [35]. As the air mass rises along the slope, the continuous rainout process creates

residual water vapor at higher elevations that is increasingly depleted in the heavy isotopic species due to preferential partitioning of the heavy isotopic species into the liquid phase relative to the vapor phase. In addition, the rise of moisture results in a decreased temperature of condensation, which further increases the fractionation between the liquid and vapor phases. Thus, the altitude effect is caused by both the evolution of the parent, condensing water vapor and the temperature of condensation. The elevation of water vapor along the slope is a driving factor of the altitude effect; therefore, the altitude effect on the windward slope is more significant than on the orographic slope, and the source of water vapor at the place where the altitude effect is significant is also relatively single.

The south side of the Himalayas is mainly under the Indian monsoon. The Himalayas act as the orographic barrier and first-order control on the regional precipitation patterns. Thus, as water vapor transported by the Indian monsoon rises along the south side of the Himalayas, both rainout and fractionation result in depletion in precipitation $\delta^{18}O$, and the $\delta^{18}O$-elevation gradient is $-0.15‰/100$ m [10]. In the orographic rain shadow of the Himalayas and the interior plateau, $\delta^{18}O$-elevation relationships are less straightforward [1,36]. Lechler et al. [1] observed that the $\delta^{18}O$-elevation gradient (-0.08‰/100 m) of the orographic rain shadow on the eastern side of the Sierra Nevada was two to three times lower than that of the orographic slope on the western side $(-0.2‰/100$ m). This climate was mainly controlled by the westerly belt in the Heihe River Basin. The Qilian Mountains' axis strikes essentially east–west and is mainly composed of a series of parallel mountains and wide valleys extending in the NWW direction. The dip direction of the slope is mainly to the north or south. The migration path of the westerly winds is almost parallel to the trend of the Qilian Mountains. Moving from west to east, the water vapor transported by the westerly wind lacks the dynamic process of climbing along the slope, which results in an insignificant precipitation $\delta^{18}O$-elevation gradient despite the large elevation change.

In addition to topographical factors affecting the evolution of stable isotopes in precipitation, water vapor serves as the material basis for precipitation, and its source and evolution also affect the variation of stable isotopes in precipitation. Mechanisms that have been proposed to explain regional variability in $\delta^{18}O$-elevation gradients primarily consider the role of atmospheric processes, including (1) mixing of moisture sources, a common process in continental interiors, (2) the role of convective storms and their associated violation of Rayleigh distillation processes in open systems, (3) the re-evaporation and recycling of $^{18}O$-enriched continental waters, and (4) the "amount effect," which accounts for drop–size–dependent isotopic equilibration times [1]. Each of these processes may act to produce precipitation $\delta^{18}O$-elevation gradients that differ from those predicted by a simple Rayleigh distillation rainout and produce an increase in the regional changes in precipitation $\delta^{18}O$-elevation gradients. Since the Heihe River Basin is located in an inland arid area of Northwest China, mixing of moisture sources, re-evaporation, and recycling of $\delta^{18}O$-enriched continental waters, and evaporative enrichment in the falling droplets beneath the cloud base are very dynamic. This results in an insignificant relationship between precipitation $\delta^{18}O$ values and elevation.

According to the atmospheric circulation around the Tibetan Plateau, the Tibetan Plateau is divided into three zones: the monsoon zone, the transition zone, and the westerlies zone [31]. The monsoon zone is generally located south of $30°$ N, the transitional zone is located between $35°$ and $30°$ N, the westerlies zone is located north of $35°$ N, and the headwaters of the Heihe River are located in the westerlies zone of the Tibetan Plateau, as shown in Figure 6. In the Indian monsoon-dominated zone, water vapor transported by the Indian monsoons is blocked by the orographic barrier, forming orographic precipitation on the south slope of the Himalayas (the windward slope). The precipitation process follows the Rayleigh fractionation process of a single water vapor source; therefore, the precipitation $\delta^{18}O$-elevation gradients in this region are significant. There was a well-defined relationship between $\delta^{18}O$ and elevation along the orographic slope $(-0.15‰/100$ m [1,10]). As the distance from the Himalayan front increased, $\delta^{18}O$-elevation relationships were less straightforward in the transitional zone due to the influence of water vapor from different sources.

In the westerly zone, the headwater regions of the Heihe River in the Qilian Mountains are located far from the Himalayan front, and a disconnect between $\delta^{18}O$ and elevation was observed. In short, the precipitation $\delta^{18}O$-elevation gradient over the Tibetan Plateau showed a gradual decrease from south to north. The precipitation $\delta^{18}O$-elevation gradient in the Qilian Mountains in the northern margin of the Tibetan Plateau was almost zero, as shown in Figure 6.

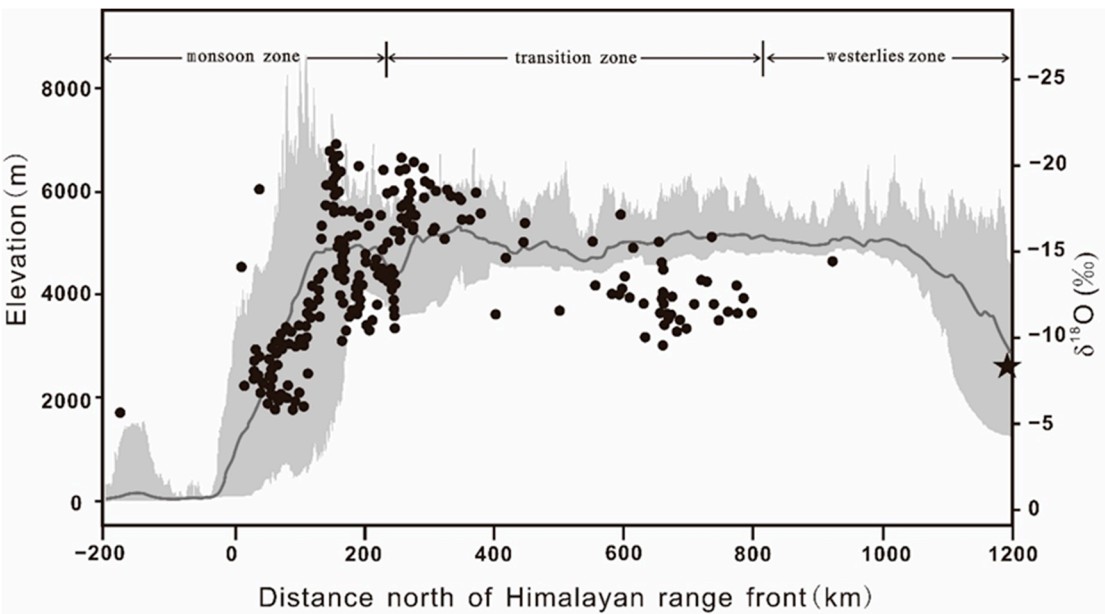

**Figure 6.** Topographic swath profiles (left y-axis) and meteoric water $\delta^{18}O$ values (right y-axis) plotted as a function of distance from the major regional orographic barrier for the Himalayan–Tibetan Plateau orogen. Upper and lower bounds of the gray swath mark the maximum and minimum elevations, respectively. Dark-gray lines mark mean elevation. Solid circles indicate $\delta^{18}O$ values, the star indicates the precipitation-weighted annual mean $\delta^{18}O$ in the upper reaches of the Heihe River Basin (compiled from Lechler et al. [1]).

## 5. Conclusions

This study demonstrated that the altitude effect in precipitation $\delta^{18}O$ between 1600–3300 m does not appear to be significant in the Qilian Mountains, and the precipitation-weighted annual mean $\delta^{18}O$ was measured to be −7.1‰. This was the result of the combined action of water vapor from different sources and orographic factors. This indicated that the spatial relationship between the geographic orientation of mountains and the main water vapor transport trajectory are important factors affecting stable isotope characteristics in precipitation. The results of this study are significant for the accuracy of estimating the paleoelevation of the Tibetan Plateau, and also indicate that the judgment of the formation area of water resources and groundwater recharge height in the Heihe River Basin based on the $\delta^{18}O$-elevation gradients in modern precipitation may need to be reconsidered.

**Author Contributions:** J.H. designed the study, analyzed the results and wrote the manuscript; W.Z. and Y.W. made editing corrections and improvements to the manuscript.

**Funding:** This research was funded by the National Natural Science Foundation of China (grant nos. 41871053 and 41690144), the Geological Investigation Project of the Chinese Geological Survey (grant nos. DD20160060) and the Chinese Academy of Sciences (CAS) 'Light of West China' Program.

**Acknowledgments:** Special thanks are given to anonymous reviewers for very helpful suggestions. Our thanks are also given to all those involved with precipitation sample collections.

**Conflicts of Interest:** The authors declare no conflict of interest.

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
