# Peer review of "The Impact of Mountain Range Geographic Orientation on the Altitude Effect of Precipitation δ18O in the Upper Reaches of the Heihe River Basin in the Qilian Mountains"

_water, doi:10.3390/w10121797_

Round 1

Reviewer 1 Report

The article “The impact of mountain range geographic orientation on the altitude effect of precipitation δ18O in the upper reaches of the Heihe River Basin in the Qilian Mountains” is well written.

The method is well described though I have a couple of questions:

1)      Line 112-113 you indicated that data were collected for period oct 2007-sept 2009. But in line 129-141 you focused on oct 2007- sept 2008

2)      Line 149-158: the linear relationship between precipitation and elevation is arguable as you may find in the article “Djebou, D.C.S., Singh, V.P. and Frauenfeld, O.W., 2014. Analysis of watershed topography effects on summer precipitation variability in the southwestern United States. Journal of Hydrology511, pp.838-849“. You may want to discuss further your section.

3)      Line 239-241 the gradual change of δ18O  in relation with the elevation could somewhat be explained by the ambient temperature gradient

In all the article is well structured and the finding is reported in an appropriate way. I, therefore, suggest it publication. 

Author Response

Point 1: Line 112-113 you indicated that data were collected for period oct 2007-sept 2009. But in line 129-141 you focused on oct 2007- sept 2008.

Response 1: Thank you for your careful work. The Yeniugou meteorological station samples between Oct 2008 and Sep 2009 were omitted, so the precipitation δ18O-elevation gradient discussion focuses only on isotopic data collected from Oct 2007 to Sep 2008 (one full year), see Line116-119. We have made an accurate statement in the new manuscript.

Point 2: Line 149-158: the linear relationship between precipitation and elevation is arguable as you may find in the article “Djebou, D.C.S., Singh, V.P. and Frauenfeld, O.W., 2014. Analysis of watershed topography effects on summer precipitation variability in the southwestern United States. Journal of Hydrology511, pp.838-849“. You may want to discuss further your section.

Response 2: Thank you very much for your instructive suggestions. According to your helpful advice, we have read this article carefully. We believe that the conclusions of this article support our views. So, we quoted this article in line 266-269. Thank you again.

Point 3: Line 239-241 the gradual change of δ18O  in relation with the elevation could somewhat be explained by the ambient temperature gradient.

Response3: Thank you for your valuable advice. Generally speaking, the stable isotope ratios (18O/16O, 2H/1H) of rainwater reflect the origin of the atmospheric vapour and the conditions of rain formation above the sampling site. Thus, the isotopic variations of rainwater during individual precipitation events allow us to assess gradual or abrupt changes in the conditions and parameters governing the rain formation. The most important parameter is temperature, which determines the extent of the atmospheric vapour condensation and the associated isotopic fractionations and rainout. So the isotope fractionation factor that defined as the ration of the two isotope ratios depends on temperature.

The variation of rain isotopic composition with altitude can be fitted by a numerical model based on a Raleigh adiabatic condensation process. The physical basis behind the Rayleigh equation that governs precipitation isotope is that isotopes are exchanged between moisture in rising air and falling droplets and precipitation. Rayleigh models have been invoked to explain the dependency on temperature of the isotopic variations of precipitation and the consequent seasonal, altitude, latitude, continentality effects. So, we did not mention the effect of the ambient temperature gradient in line 260-261 in the new manuscript.

Reviewer 2 Report

This paper compared the results of estimating the annual precipitation with the altitude using the δ18O-elevation gradients method and the actual precipitation values in the study area.

The results of the comparison show that the precipitation δ18O-elevation gradients method results are not significant compared to the actual precipitation results in the study area.

The results of this study suggest that the precipitation δ18O-elevation gradients method is not applicable to all regions and may differ from actual results depending on various geographical characteristics and climate patterns.

This study suggests that there is a need to review the results of regions that have determined water resources and groundwater recharge heights based on the relevant research methods.

However, there are some things that need to be corrected.

The Figure 2’s label and legend are unclear. Increase the size of each figure.

Table 1 does not clearly indicate what it means. Please explain further.

Line 153-156: You write "they found a linear relationship." However, it is hard to understand that R2= 0.22 result is a linear relationship. A further explanation is needed.

Line 215:  ->  require correction

Author Response

Point 1: The Figure 2’s label and legend are unclear. Increase the size of each figure

Response 1: Thank you very much. According to your comment, we have increased the size of the figure in the new manuscript.

Point 2: Table 1 does not clearly indicate what it means. Please explain further

Response 2: Thank you for your instructive suggestions. According to your help advice, we have explained the abbreviations in table1 and table2.

Point 3: Line 153-156: You write "they found a linear relationship." However, it is hard to understand that R2= 0.22 result is a linear relationship. A further explanation is needed.

Response3: Thank you for your help advice. In this article quoted by us, the authors think that “they found a linear relationship” though R2= 0.22. We are very sorry for our inaccurate statement. We have corrected it in the new manuscript. We have the same opinion as you, and consider that the regressions between precipitation-weighted mean δ18O and elevation showed weak correlation with an R2 value of 0.22, see line 174-155.

Point 4: ->  require correction

Response4: Thank you for your carefully reading. We are sorry for this mistake. And We have corrected it in the new manuscript.

Water EISSN 2073-4441 Published by MDPI AG, Basel, Switzerland RSS E-Mail Table of Contents Alert
Back to Top